# Responsible Use of Artificial Intelligence in Dentistry: Survey on Dentists’ and Final-Year Undergraduates’ Perspectives

**DOI:** 10.3390/healthcare11101480

**Published:** 2023-05-19

**Authors:** Jelena Roganović, Miroslav Radenković, Biljana Miličić

**Affiliations:** 1Department of Pharmacology in Dentistry, School of Dental Medicine, University of Belgrade, 11000 Belgrade, Serbia; 2Department of Pharmacology, Clinical Pharmacology and Toxicology, Faculty of Medicine, University of Belgrade, 11000 Belgrade, Serbia; miroslav.radenkovic@med.bg.ac.rs; 3Department of Medical Statistics and Informatics, School of Dental Medicine, University of Belgrade, 11000 Belgrade, Serbia; biljana.milicic@stomf.bg.ac.rs

**Keywords:** artificial intelligence, ethics, dentistry, conflict of interest, accountability, bias, regulatory policy

## Abstract

The introduction of artificial intelligence (AI)-based dental applications into clinical practice could play a significant role in improving diagnostic accuracy and reforming dental care, but its implementation relies on the readiness of dentists, as well as the health system, to adopt it in everyday practice. A cross-sectional anonymous online survey was conducted among experienced dentists and final-year undergraduate students from the School of Dental Medicine at the University of Belgrade (n = 281) in order to investigate their current perspectives and readiness to accept AI into practice. Responders (n = 193) in the present survey, especially final-year undergraduates (n = 76), showed a lack of knowledge about AI (only 7.9% of them were familiar with AI use) and were skeptical (only 34% of them believed that AI should be used), and the underlying reasons, as shown by logistic regression analyses, were a lack of knowledge about the AI technology associated with a fear of being replaced by AI, as well as a lack of regulatory policy. Female dentists perceived ethical issues more significantly than men regarding AI implementation in the practice. The present results encourage an ethical debate on education/training and regulatory policies for AI as a prerequisite for regular AI use in dental practice.

## 1. Introduction

With its strong ability to integrate large sets of clinical data, artificial intelligence (AI) has shown potential as a useful medical tool in diagnosis [1] and clinical decision making [2]. In dentistry, various AI-based software-type algorithms are used, which are expected to improve the accuracy of dental diagnosis by overcoming the limitations of the current radiographic techniques and improving the quality of radiological images [3]. Currently, many dentists are awaiting the integration of AI systems into diagnostics, prognostics and dental treatment; however, as AI system adoption expands throughout the dental sector, questions about legal and ethical dilemmas when using this technology become ever more pertinent [4,5,6]. AI technology will and already is affecting the educational sector, as seen during the COVID-19 pandemic, aiming at providing a personalized learning approach via an interactive learning experience [7]. However, many educational institutions face challenges with the effective adoption of AI into their teaching practice due to a lack of teachers’ training in AI, the high costs of AI software and ethical concerns [8,9]. The recent introduction of the new, powerful AI-driven language model, ChatGPT-4, immediately showed its potential in enabling students to embrace even complex scientific concepts, but it also highlighted a number of legal and ethical concerns [10].

As an example, let us imagine a patient for whom the dentist recommends tooth extraction as an orthodontic treatment, and the dentist’s recommendation is based on the analysis of an artificial intelligence (AI) system, which recommends this treatment plan based on a cephalometric analysis. The decision to extract teeth for orthodontic treatment is sensitive and difficult because it is based mainly on the practitioner’s experience, and it is irreversible [11]. What if, due to an incorrect decision based on the AI’s knowledge of tooth extraction, undesirable results occur, such as the failure of anchorage control, abnormal inclination of the anterior teeth or improper occlusion [12], and the orthodontic treatment cannot be finished? Who is to blame? Laws and legal frameworks are not specifically defined for AI use in dentistry. The currently defined medical liability in legal frameworks is inadequate and thus may not encourage safe use of AI in clinical decision making [13]. It can be stated with certainty that AI use in clinical decision making needs dentist surveillance, and the role of dental practitioners is crucial in preventing dental complications, as well as in reviewing AI systems [14,15]. Thus, our aim was to investigate dentists’ knowledge and perspectives towards AI systems’ use in dentistry, as it is of importance for AI system adoption in the dental practice.

## 2. Materials and Methods

The manuscript was prepared according to the Checklist for Reporting Of Survey Studies (CROSS) (Appendix A). A self-administered, 25-item questionnaire with closed-ended questions in the Serbian language was used, prepared based on a review of the literature assessing attitudes toward AI and dental ethics, as well as on informal discussions with available practicing dentists and health ethics researchers [16,17]. As these discussions revealed that there was a lack of knowledge about the subject, we used a 3-point Likert-type scale [18] to obtain a simple and straightforward response from participants in a “forced-choice” response format. The purpose was to encourage participants to carefully consider their responses and to reduce the response bias that can occur when participants always select the neutral option. Based on the assumption that participants were able to compare the items and make relative judgments about them, even if they may not have been able to provide precise or accurate ratings, we used the rank order scale.

This study was conducted in accordance with the code of ethics of the World Medical Association (Declaration of Helsinki), and the investigation was approved by the Ethical Committee of the School of Dental Medicine (number 36/7, 2022). This cross-sectional online survey used convenience sampling and was conducted between February and April 2022. The survey aimed to assess dentists’ and final-year students’ familiarity with and attitudes toward AI use in dentistry, without any specific hypothesis testing. Thus, sample size estimation and power calculation were waived. Invitations to participate in the study were sent via e-mails to experienced dentists working at the university (n = 151) and students in the final year of their undergraduate studies (n = 130). Thus, participants represented a group of dentists with various levels of education and skills, reflecting the dental community in Serbia. Participation was voluntary and anonymous and only the research team had access to the survey data, which were kept in a secure location. E-mail verification was used to prevent the “multiple participation” of participants. As the research project was clearly described in the note accompanying the questionnaire, the completion of the questionnaire by the respondents was regarded as their consent to participation in the study. The questionnaire consisted of four parts that aimed to provide information on demographic data, previous experience with the application of artificial intelligence in dental practice, attitudes towards artificial intelligence and responsibility issues related to the application of artificial intelligence. The average time to complete the questionnaire was 10–12 min. The data were collected and managed using SPSS Version 28.0 (IBM, Armonk, NY, USA). All our data were categorical. To address non-response error in the analysis of survey data, we employed imputation. Descriptive statistics were presented as frequencies, and proportions were used for discrete measures. A two-sided chi-square test was used to calculate group differences. Univariate and multivariate logistic regression were used to investigate whether the participants’ gender, working experience or academic (PhD) and clinical expertise (specialization) were significantly associated with or predicted questionable attitudes and perspectives. A *p*-value of less than 0.05 was considered significant.

## 3. Results

### 3.1. Participants

From a total of 281 invitations sent, 193 responses were obtained, representing a response rate of 68.7%: seventy-six (58.5%) from students and 117 (77.5%) from experienced dentists. All students were under 30 years of age, with a male/female ratio of 16/60. The gender distribution was similar in the group of experienced dentists (39/77, *p* = 0.06). The demographic data of the participants are presented in Table 1.

### 3.2. Previous Experience with and Attitudes toward AI Use

As a measurement of the internal consistency of the questionnaire items in two areas, attitudes and perspectives (items placed in these groups, the first measuring the familiarity with AI and the second measuring attitudes towards ethical issues related to AI, are presented in Table 2), Cronbach’s alpha was estimated, and it was 0.76 for attitudes and 0.78 for perspectives. It was obtained by computing the correlation matrix separately for each group of items based on the sample of 30 participants.

The results showed that 27.1% of participants were not familiar with AI use in dentistry/medicine, and only 10.9% of them were currently using or were familiar with AI dental applications. Although the majority were of the opinion that AI use would improve the efficiency and quality of a dentist’s work, less then half of them had an affirmative response to whether AI should be used in the dental practice. Regarding ethical issues, the most significant was considered the lack of regulatory policies, followed by the issue of no specified data protection measures and privacy issues (Table 2). Moreover, women were more aware of the significance of the ethical issues associated with AI use in dental practice than men (Figure 1).

According to the pre-investigation experience and the principle of the rank order scale, the basis for the classification of the three ranks was determined. The grading “yes” or “very significant” indicated a level of ≥80%; the grading “Partly” or “significant” indicated an acceptance level of 20% and 79% (including 20% and 79%); the grading “No” and “non-significant“ indicated an acceptance level of ≤19% [19].

By using univariate logistic regression analysis, we found that a lack of working experience (being a student) was significantly associated with a lack of knowledge regarding AI use in medicine and dentistry (only 7.9% of them were familiar with AI use in medicine/dentistry), concerns about being replaced by AI technology (8% replied “yes” and 35% replied “partly”) and doubt about AI use in dentistry (only 34% of them felt that AI should be used in the dental practice). Students were more concerned about the lack of regulatory policy when using AI than experienced dentists (77% of them, in comparison to 65% of experienced dentists, considered this matter highly significant). At the same time, having a PhD or specialization was shown to be associated with the level of knowledge of AI use in dentistry/medicine. Females considered ethical issues to be more significant than men (Table 3).

Results of the multivariate logistic regression analysis revealed that being a student was predictive factor of a lack of knowledge, a fear of dentists being replaced by AI and skepticism concerning AI use in dental practice (Table 4).

Regarding responsibility, most of the participants reported that the dentist would be accountable if harm to the patient occurred due to AI use and the AI developer would be accountable for false AI recommendations due to algorithmic bias, while the Ministry of Health would be regarded as responsible for AI accuracy monitoring (Figure 2).

Considering the issue of data management, the highest number of participants agreed that the Ministry of Health should be responsible for data protection from misuse and cybercrime, and collected data “belong“ to the health institution/medical worker using the AI. Almost all participants agreed that informed consent was necessary for data management (Figure 3).

## 4. Discussion

The present survey considered dentists’ perceived responsibility while using AI in terms of the dentist’s experience and qualifications and the prudent use of AI, and it suggested that an ethical debate on education and regulatory policy regarding AI is a prerequisite for AI application in dentistry.

### 4.1. Dentist Qualification

The present results revealed that only 10.9% of survey responders were currently using or were familiar with AI dental applications. Similar results were obtained among German dentists, where most of them reported that their knowledge about AI was average and only 6.3% reported that it was excellent [20]. However, a survey conducted among dental students in Turkey showed that 48.4% of dental students had basic knowledge of AI technologies, while 10.6% possessed no information about AI [21].

The results of the present survey show that working experience and having a specialization/PhD degree are associated with greater motivation to use and knowledge about AI. Moreover, undergraduates are skeptical about whether they should use AI in the practice at all. Such negative attitudes regarding AI use will more likely slow the adoption of AI in the practice, since both patients and health systems are sensitive to the dentist’s reaction. Likewise, one’s attitudes towards AI use in dentistry may influence which issue of AI use should be addressed first and before the implementation of the technology into practice. In particular, the skepticism about AI use in the present study is associated with “anxiety” regarding a lack of regulatory policies. Thus, if one is skeptical about AI use, then one favors a more robust regulatory policy to be implemented. The current unpreparedness of the University of Belgrade’s dental community for AI use underlies the evident lack of basic and continuing education regarding this subject, as well as the fear of dentists being replaced by AI, both factors shown to be important previously [22]. However, the present survey reveals that a lack of regulatory policy represents an important factor also, since it could leave both dentists and patients exposed to legal uncertainty when using AI.

One goal of AI use is enabling the general medical practitioner/dentist to achieve a higher level of sophistication in diagnosis or treatment, similar to that of a specialist, by using AI devices as support in making decisions. Under such circumstances, a lack of experience/qualification in a dentist may represent an obstacle and a matter of responsibility and liability [23]: the general dentist is unqualified to perform at the level of a specialist. In line with this, educational programs within dental studies, as well as carefully planned and conducted AI training, are a necessity in achieving responsible AI use in the dental practice. The majority of our participants felt that both undergraduate and postgraduate studies should offer insights into AI-guided technology.

### 4.2. Prudent Use of AI in Dentistry

#### 4.2.1. Regulatory Policy Should Consider AI Use and Accountability if Harm to the Patient Occurs

In this survey, dentists regarded the lack of regulatory policy as the AI-associated ethical issue of the highest significance. Notably, women rated ethical issues to be more significant than men. Indeed, previous studies showed that women’s behavior is more consistent with their internal moral principles and that they are more likely to experience moral-related emotions such as guilt and shame than men [24,25], which can influence their moral decision making. A regulatory policy should include information on whether the AI was reviewed by the FDA or another regulator, but also how and when doctors should use AI: whether they need to follow the AI recommendation always or overrule it in situations in which they disagree with the recommendation, for example. Notably, there could be a difference in the regulation of the mode of implementation depending on whether the AI recommendation was merely “informational” or whether the AI should be treated as a major clinical decision tool [26]. Under the latter circumstances, what should the dentist do if he/she has the opposite recommendation? Should he/she reject the recommendation and record the reasons or consult a colleague or a board of experts and then make a decision? Moreover, there is also ethical complexity regarding the multiple players in a healthcare system. Namely, even if dentist decides not to act according to the AI recommendation, one could imagine pressure from other players in the chain of healthcare, such as from the insurance company (for example, if the recommendation is to use AI, they may cover only treatment according to the AI-based decision). Currently, AI tools in dentistry are designed as support tools in clinical decision making, and the results of the present survey show that dentists are aware of their own accountability when it comes to the occurrence of medical errors and harm to the patient while using AI systems. Indeed, given the lack of case law on medical doctor/dentist use of AI, it is suggested that doctor is accountable for errors resulting from AI use: the dentist may be liable for failing to critically evaluate the AI recommendations, such as misusing an AI decision support tool as a primary diagnostic tool, or implementing an improper AI system in the dental practice. In general, health system authorities (Ministry of Health or health facilities) may be liable for failing to provide training, updates, support, maintenance or equipment for an AI algorithm [14].

As with other drugs or medical devices, the adoption of AI in practice should follow approval by regulatory agencies [27]. Namely, if we consider AI as another “member of the dental team”, then it is necessary to have information about its “expertise”. In its latest update in 2022, the FDA authorized over 500 AI-enabled medical devices [28], mainly in the radiology sector. For the dental domain, the FDA approved *VideaHealth’s* AI algorithm, which has been shown in clinical trials to lower the rate of missed cavities by more than 40%, while also reducing incorrect caries diagnoses by about 15%; *Overjet’s Dental Assist* software, which uses deep learning AI to automatically measure bone loss in X-rays, reducing the amount of time required for dentists to diagnose and begin treating periodontal disease; and the *Pearl’s Second Opinion* solution, which relies on computer vision AI to identify conditions such as cavities, tartar and inflammation, while also highlighting crowns, fillings, root canals, bridges and implants. However, the regulatory practices of the FDA have been questioned recently, where authors have emphasized that there are no established best practices for the evaluation of commercially available algorithms to ensure their reliability and safety. Based on publicly available information on FDA-approved devices, the authors concluded that almost all of the AI devices approved underwent only retrospective studies, while a substantial proportion of approved devices have been evaluated only at a small number of sites, suggesting limited geographic diversity [29].

#### 4.2.2. Performance of AI across Representative Populations and How to Avoid Algorithmic Bias

In this study, the dentists identified doctors, health systems and algorithm developers as subjects within different, overlapping areas of accountability for AI use. While doctors and health systems were considered accountable under AI-use-associated malpractice, algorithm designers were considered accountable for failures regarding product safety and reliability. There is overall concern that in medical/dental AI, deficiencies in the representativeness of the data sets used to train the system may result in inadequate performance for different races, ages, genders, etc. For example, regarding the previously mentioned AI-based recommendation system for tooth extraction for orthodontic treatment, it has been shown that there are variations in cephalometric landmarks between Caucasian men and African American men [30]. If the training data lacked sufficient data on African American men, the algorithm used for cephalometric analysis would likely give treatment recommendations that are not appropriate for the African American population. What should one do in such cases? One possibility would be not to use AI. Another might be to limit the use of this supportive tool only to specialists in orthodontics. Thus, when it comes to liability, algorithm developers could be accountable for injuries that result from poor design, failure to warn about risks or manufacturing defects. On the other hand, the health system could be liable for injuries resulting from an inaccurate AI system since it recommends which AI systems should and should not be used in clinical practice; thus, failure to conform to the standard of care for each particular group may be the result of an incorrect decision [14].

#### 4.2.3. Bias in Professional Judgment Due to Conflict of Interest

In regard to identifying doctors as accountable when using AI, it is noteworthy that dentist participation sometimes may create bias that could adversely influence the decision. As an example, when a doctor is involved in the development of AI applications for dental use, as a founder or board member in companies engaged in AI dental applications, a conflict of interest may arise. The dentist must disclose their personal interests unrelated to the patient’s health, whether research or economic, that may affect his/her professional judgment. The liability issue exists due to the fact that the financial motive (to commercialize a developed AI application) may compromise the reasoning of the dentist when it comes to decision making [31]. If the dentist has a conflict of interest and favors his/her self-interest, it may violate his/her binding legal duties, such as fiduciary duties. Namely, the legal system recognizes the doctor’s duties to patients as fiduciary duties, which oblige the doctor to act in a manner to protect not only the patients’ health but also their overall interests from the doctor’s self-interest [32]. Under such circumstances, the resolution of a conflict of interest could be achieved by preventing the dentist from being directly involved, or, when prevention is not necessary or not possible, by disclosing it. A feasible strategy for a dentist’s disclosure could be to inform patients, as well as the general public, before the treatment or research, and allow the patient to decide whether or not such interests are acceptable to them [32].

#### 4.2.4. Data Management Control

Data management is one of the fundamental ethical issues associated with acquiring, assessing and storing data. Dentists should use the collected data only to improve the patients’ health, as well as to improve the dental practice. It is unethical to use data in ways that may harm or in any way adversely affect patients. One of the cornerstones of data management control is data privacy. According to the General Data Protection Regulation (GDPR), patients have the right to know about any personal data that the dentist holds about them and how she/he plans to use it; likewise, they have the right to restrict access or ask her/him to stop processing or delete their data [33]. Therefore, in order to ensure data security, the dentist should limit the amount of personal data collected or delete data that are no longer needed; pseudonymize or anonymize personal data whenever possible; ensure e-mail security and device encryption; and notify the patients if a data breach occurs [33]. The results of the present questionnaire show that dentists consider the ethical problem of the lack of clearly defined guidelines on the protection of patient data during AI use to be very significant Moreover, they perceive the Ministry of Health as the most responsible for solving this problem. Likewise, the Ministry of Health is perceived as the most authoritative in protecting data from cybercrime. Students and dentists from the University of Belgrade unequivocally agreed that the written consent of the patient is necessary for the collection and processing of data during the application of AI.

Thus, the responsible use of AI in dentistry should be focused on using AI to promote the quality of oral and systemic health by:-using legitimately approved and regulated AI software;-engaging in obligatory education and training in AI use and the continuous monitoring of AI-based system performance in the particular group of patients;-protecting patients and implementing vigorous data management control;-extensively informing patients if there is any possibility that the AI system is being used for reasons other than to improve patient health, such as when a dentist or health facility has a conflict of interest.

While the present study was the first one to consider the debate regarding ethical concerns related to the use of AI, it was performed on a geographically limited number of participants and lacked sample size estimation and sensitivity analyses, which represent the limitations of the study. Thus, further research should be more robust and applicable to other populations and contexts and encompass ethical considerations arising from the use of other potential digital clinical and educational tools, such as the application of augmented reality, virtual reality and mixed reality, in dentistry.

## 5. Conclusions

The skepticism about AI use shown by the dental community in the present study is associated with the evident lack of basic and continuing education regarding this subject, the fear or risk that AI will replace dentists and “anxiety” regarding the lack of regulatory policies. Female dentists perceived ethical issues to be more significant than men, while the lack of regulatory policies was considered the AI-associated ethical issue of the highest significance. The present results encourage the ethical debate on education/training and regulatory policies as a prerequisite for regular AI use in dental practice.

## Figures and Tables

**Figure 1 healthcare-11-01480-f001:**
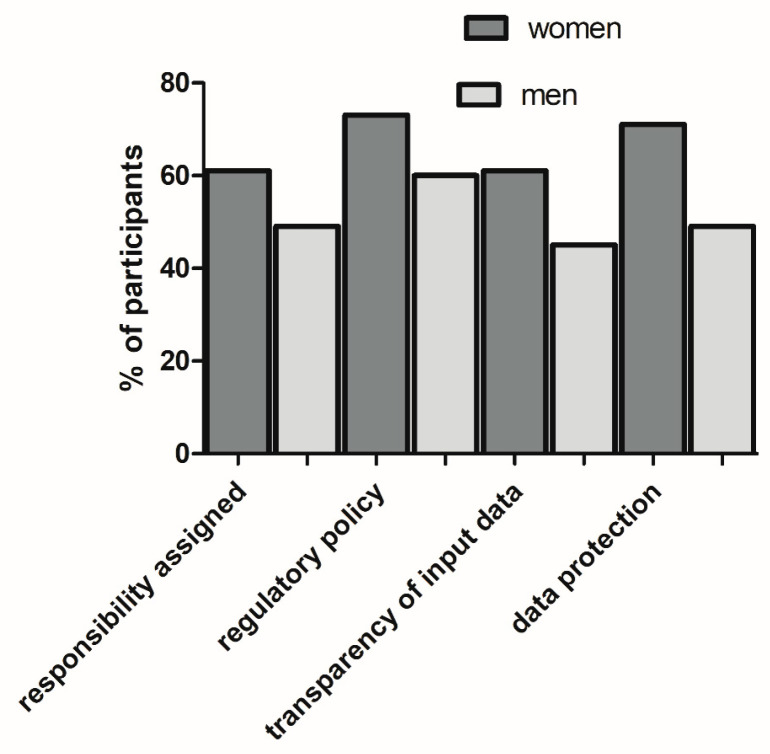
Gender differences in perspectives on the significance of different ethical issues associated with AI use. Bars represent the number of participants perceiving each ethical issue as very significant.

**Figure 2 healthcare-11-01480-f002:**
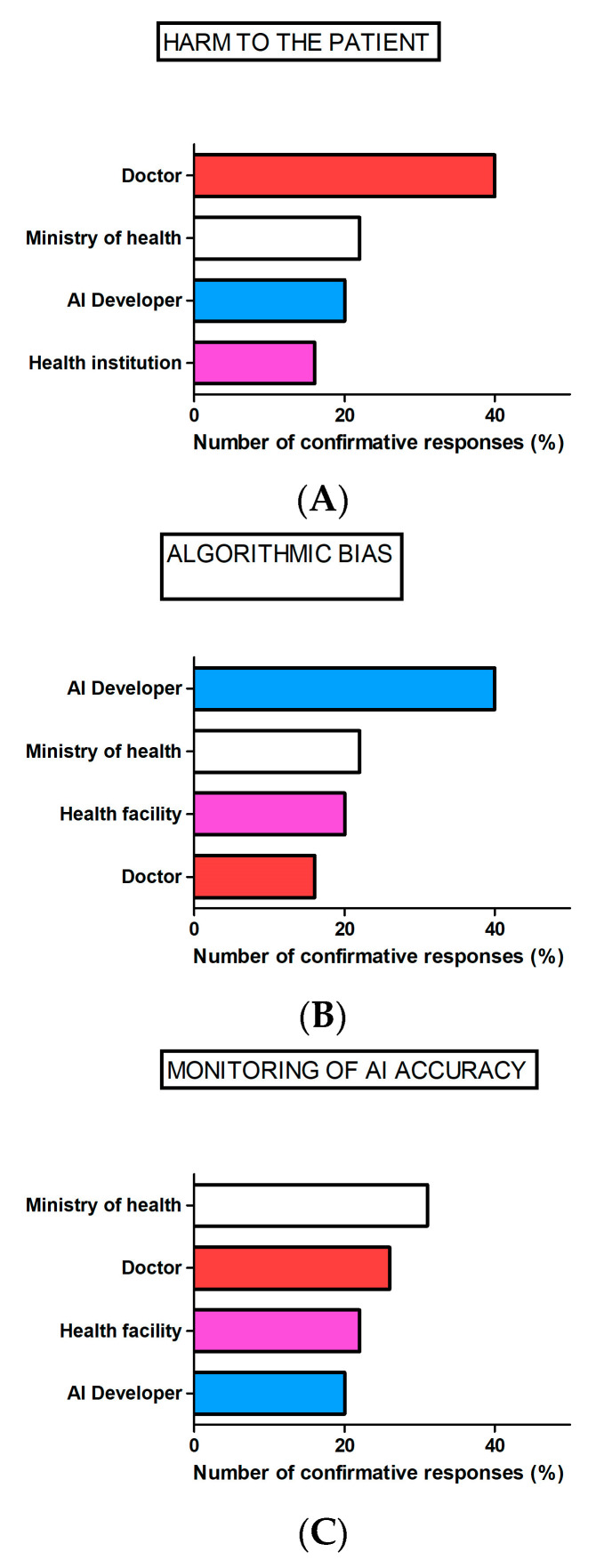
Dentists’ perceived responsibility when using AI in the dental practice. Bars represent the number of participants responding to the following questions: WHO SHOULD BE ACCOUNTABLE IF HARM TO THE PATIENT OCCURS DUE TO FALSE OR BIASED AI-BASED DECISION? (**A**), WHO SHOULD BE ACCOUNTABLE FOR FALSE AI-BASED DECISION DUE TO algorithmic BIAS? (**B**), WHO SHOULD TAKE RESPONSIBILITY FOR MONITORING OF AI-BASED SOFTWARE ACCURRACY? (**C**).

**Figure 3 healthcare-11-01480-f003:**
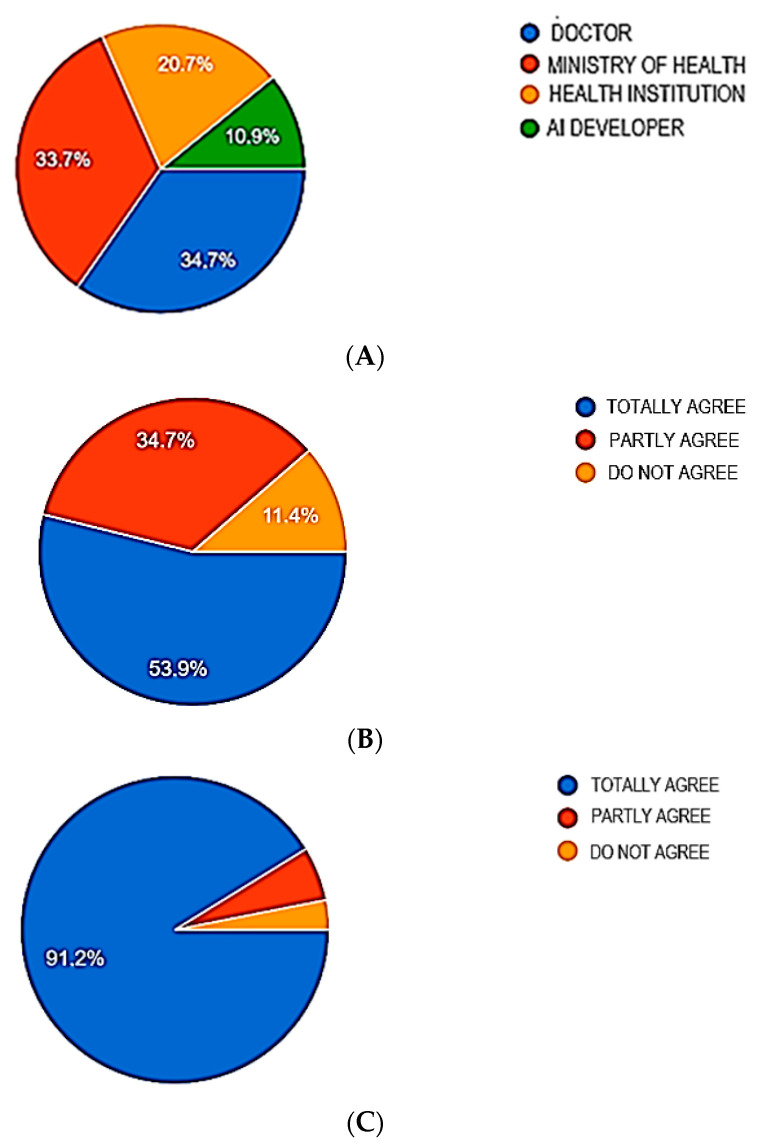
Dentists’ perceptions regarding data management control. Pie graphs represent the number of participants responding to the following questions: WHO SHOULD TAKE RESPONSIBILITY FOR PROTECTION OF AI-DERIVED DATA FROM MISUSE/CYBERCRIMINAL? (**A**), THE DATA COLLECTED USING AI SHOULD BELONG TO The health institution/medical worker which uses AI? (**B**), AN INFORMED CONSENT IS NEEDED FOR COLLECTION, ANALYSES AND SHARING AI-DERIVED DATA? (**C**).

**Table 1 healthcare-11-01480-t001:** Demographic data of the survey participants.

Characteristic	Participants, Number (Percentage)
**Age group (year)**	
<30	96 (49.7%)
30–45	57 (29.5%)
>45	40 (20.7%)
**Gender**	
Male	55 (28.5%)
Female	138 (71.5%)
**Work experience (years)**	
Student	76 (39.4%)
<10	56 (29.0%)
>10	61 (31.6%)
**Postgraduate degree**	
PhD	39 (20.4%)
Specialization	61 (32.3%)
none	94 (48.7%)

**Table 2 healthcare-11-01480-t002:** Attitudes toward AI use in dentistry and perspectives toward ethical issues associated with AI use.

QUESTION—ATTITUDES	ANSWER—YES (Number)	ANSWER—PARTLY (Number)	ANSWER—NO (Number)
ARE YOU FAMILIAR WITH THE AI APPLICATIONS USE IN HEALTH CARE?	28 (14.6%)	112 (58.3%)	52 (27.1%)
DO YOU USE OR FAMILIAR WITH USAGE OF CURRENT APPLICATIONS OF AI-BASED DENTAL SOFTWARE (ORCA Dental AI, Denti AI, VideaHealth, Pearl, Glidewell.io, Smilecloud, DentalXrai Pro, Dental Analytics, Dental monitoring, AssistDent etc.)?	21 (10.9%)	83 (43.0%)	89 (46.1%)
DO YOU THINK AI COULD REPLACE DENTISTS IN A DENTAL PRACTICE?	7 (3.6%)	62 (32.3%)	123 (64.1%)
DO YOU THINK AI COULD IMPROVE THE QUALITY OF DENTAL WORK IN A DENTAL PRACTICE?	99 (51.3%)	78 (40.4%)	16 (8.3%)
DO YOU THINK AI COULD IMPROVE THE EFFICACY OF DENTISTS IN A DENTAL PRACTICE?	110 (57.3%)	70 (36.5%)	12 (6.2%)
DO YOU THINK AI SHOULD BE USED IN A DENTAL PRACTICE?	93 (48.7%)	86 (45.0%)	12 (6.3%)
AI APPLICATIONS IN DENTISTRY SHOULD BE INCORPORATED IN UNDERGRADUATE DENTAL CURRICULA?	88 (45.8%)	77 (40.1%)	27 (14.1%)
AI APPLICATIONS IN DENTISTRY SHOULD BE INCORPORATED IN POSTGRADUATE DENTAL CURRICULA?	130 (68.1%)	51 (26.7%)	10 (5.2%)
**QUESTION—PERSPECTIVES**	ANSWER—VERY SIGNIFICANT	ANSWER—SIGNIFICANT	ANSWER—NON-SIGNIFICANT
HOW DO YOU ESTIMATE THE SIGNIFICANCE OF THE FOLLOWING ETHICAL ISSUE: The lack of assigned responsibility for consequences of AI-based decision	112 (58.0%)	72 (37.3%)	9 (4.7%)
HOW DO YOU ESTIMATE THE SIGNIFICANCE OF THE FOLLOWING ETHICAL ISSUE: The lack of transparent and representative data used for AI software development	110 (57.0%)	72 (37.3%)	11 (5.7%)
HOW DO YOU ESTIMATE THE SIGNIFICANCE OF THE FOLLOWING ETHICAL ISSUE: The lack of regulatory policy which governs AI use	134 (69.4%)	52 (26.9%)	7 (3.7%)
HOW DO YOU ESTIMATE THE SIGNIFICANCE OF THE FOLLOWING ETHICAL ISSUE: The data protection and privacy issues associated with AI use not specified	125 (64.8%)	58 (30.1%)	10 (5.1%)

**Table 3 healthcare-11-01480-t003:** Explanatory factors associated with the analyzed attitudes and perspectives (results from univariate logistic regression analyses).

ATITUDES AND PERSPECTIVES	Significant Association with	Exp (B)	95%CI	*p*-Value
ARE YOU FAMILIAR WITH THE AI APPLICATIONS USE IN HEALTH CARE?	Being student Holding PhD Having dental specialization	0.575 6.671 2.431	0.35–0.92 3.21–13.86 1.43–4.12	*p* = 0.023 *p* < 0.001 *p* < 0.001
DO YOU USE OR FAMILIAR WITH USAGE OF CURRENT APPLICATIONS OF AI-BASED DENTAL SOFTWARE?	Holding PhD Having dental specialization	3.598 1.794	2.03–6.34 1.13–2.85	*p* < 0.001 *p* = 0.013
DO YOU THINK AI COULD REPLACE DENTISTS IN A DENTAL PRACTICE?	Being student	1.763	1.04–2.96	*p* = 0.033
DO YOU THINK AI SHOULD BE USED IN A DENTAL PRACTICE?	Being student	0.518	0.31–0.84	*p* = 0.009
HOW DO YOU ESTIMATE THE SIGNIFICANCE OF THE FOLLOWING ETHICAL ISSUE: The lack of assigned accountability for consequences of AI-based decision	Being female	0.554	0.32–0.93	*p* = 0.027
HOW DO YOU ESTIMATE THE SIGNIFICANCE OF THE FOLLOWING ETHICAL ISSUE: The lack of transparent and representative data used for AI software development	Being female	0.506	0.30–0.84	*p* = 0.009
HOW DO YOU ESTIMATE THE SIGNIFICANCE OF THE FOLLOWING ETHICAL ISSUE: The lack of regulatory policy which governs AI use	Being female Being student	0.446 1.743	0.25–0.77 0.98–3.09	*p* = 0.004 *p* = 0.047
HOW DO YOU ESTIMATE THE SIGNIFICANCE OF THE FOLLOWING ETHICAL ISSUE: The data protection and privacy issues associated with AI use not specified	Being female	0.384	0.22–0.65	*p* < 0.001
WHO SHOULD BE ACCOUNTABLE FOR FALSE AI-BASED DECISION DUE TO ALGORITHMIC BIAS?	Being female	0.715	0.53–0.95	*p* = 0.02

**Table 4 healthcare-11-01480-t004:** Explanatory factors predicting the analyzed attitudes and perspectives (results from multivariate logistic regression analysis).

ATITUDES	Significantly Associated with (Predictor)	Exp (B)	95%CI	*p* Value
ARE YOU FAMILIAR WITH THE AI APPLICATIONS USE IN HEALTH CARE?	Being student	0.556	0.34–0.98	*p* = 0.03
DO YOU THINK AI COULD REPLACE DENTISTS IN A DENTAL PRACTICE?	Being student	1.913	1.06–3.46	*p* = 0.03
DO YOU THINK AI SHOULD BE USED IN A DENTAL PRACTICE?	Being student	0.510	0.29–0.87	*p* = 0.01
ARE YOU FAMILIAR WITH THE AI APPLICATIONS USE IN HEALTH CARE?	Holding PhD	5.49	2.32–13.02	*p* < 0.001

## Data Availability

The data presented in this study are available on request from the corresponding author.

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
