# Peer review of "Responsible Use of Artificial Intelligence in Dentistry: Survey on Dentists’ and Final-Year Undergraduates’ Perspectives"

_healthcare, 2023, doi:10.3390/healthcare11101480_

Round 1

Reviewer 1 Report

Dear Authors, 

I found this work impactful and fit well with in the scope of this journal. The manuscript needs some minor improvements; there are a few suggestions that authors may consider to improve it further:

The use of the English language is reasonable, however, there are a number of punctuation and grammatical errors; that should be corrected and rephrased using academic English for a better flow of text for reader.

Abstract: is precisely written, and the aim of the study is mentioned. Please include some more information about the results/finding to enhance the impact of this section.

The purpose of this study and the methodology are not well described in the abstract. What are the groups into which the patients are divided? What outcomes were assessed? What results have been achieved?

Introduction- this section is short and should be more focused on the topic in question.

Materials and methods- this section is well organized, 

Please, describe this section  meticulously as this is very important for the readers. 

Furthermore, In discussion, more studies in context should be included; as there is little support of literature from the previous studies. I would suggest this paper: 

"Clinical Applications of the Algorithm “Pipeline Advanced Contrast Enhancement (Pace)” in Dental Radiology" 

I believe that your manuscript would have much more relevance after suggested improvements.

Author Response

Thank you for the suggestions which improved our manuscript.

I found this work impactful and fit well with in the scope of this journal. The manuscript needs some minor improvements; there are a few suggestions that authors may consider to improve it further:

The use of the English language is reasonable, however, there are a number of punctuation and grammatical errors; that should be corrected and rephrased using academic English for a better flow of text for reader.

Abstract: is precisely written, and the aim of the study is mentioned. Please include some more information about the results/finding to enhance the impact of this section.

 Accordingly, we made corrections in Abstract and native speaker has gave suggestions to improve English.

The purpose of this study and the methodology are not well described in the abstract. What are the groups into which the patients are divided? What outcomes were assessed? What results have been achieved?

We added as much information as we could into Abstract, due to limited number of words allowed.

However, we rephrased Methodology section (highlighted in text)

Introduction- this section is short and should be more focused on the topic in question.

Accordingly, we expanded the Introduction and added new references.

Materials and methods- this section is well organized, 

Please, describe this section  meticulously as this is very important for the readers. 

Accordingly, we rephrased and expanded Methodology section according to Checklist for Reporting Of Survey Studies (CROSS) which is included as supplementary file.

 Furthermore, In discussion, more studies in context should be included; as there is little support of literature from the previous studies. I would suggest this paper:  

"Clinical Applications of the Algorithm “Pipeline Advanced Contrast Enhancement (Pace)” in Dental Radiology" 

We expanded the Discussion accordingly, by comparing present results with literature. The suggested reference was cited in the Introduction.

Reviewer 2 Report

The paper “Responsible use of Artificial intelligence in dentistry: dentists’ perspectives” reports on the results of the online survey conducted among experienced dentists and final year-undergraduate students in order to investigate their current perspectives and readiness to accept AI into practice. The topic is highly relevant and popular; the article is interesting for the readers. However, some comments should be addressed.

General note

Please, revise the manuscript according to Checklist for Reporting Of Survey Studies (CROSS). It is advisable to include a checklist to a supplementary materials.

Abstract

The results should be presented with at least some percentages instead of general phrases.

Instead of the concluding phrase “Present article resumed the dentists’s percieved responsibility while using AI through the lens of: the dentist experience and qualification, prudent use of AI, and existence of conflict of interest.” there should be a brief conclusion on the results of the survey. 

Materials and methods

How was the questionnaire distributed? Was it web-based? In which language was the questionnaire?

How many questions were in each section? What was the average time to complete the questionnaire?

Please, provide the results of minimum sample size calculation.

Results

Demographic data for the students and experienced dentists  should be presented separately separately in the table 1.

Please, provide the calculation of the response rate for the two study groups.

Lines 83-84 should be explained in materials and methods section: how did you calculate Cronbah’s Alfa (between which measures?), which sample was used for this purpose

Lines 96-100 should be moved to materials and methods section.

“According to the pre-investigation experience and the principle of the rank order scale, the basis for the classification of the three ranks is determined.” This should be further explained and the “principle of the rank order scale” should be referenced.

Table 2: The scale including “very significant”, “significant”, and “non-significant” contains 2 positive and 1 negative answer. Why it was used instead of a traditional Likert scale? This should be justified in the discussion.

Table 3. It is better to provide exact p-values.

Figure 3 is confusing. Its idea should be explained in a clearer way.

Discussion: 

Discussion should focus on actually discussing the results of the study. The part “Prudent use of AI in dentistry” represents a narrative review and the authors view on how ai-based systems should be used.

There is a great number of surveys on the perception of doctors and students towards artificial intelligence. These surveys and their results should be compared with the present survey in the discussion.

The limitations of the survey should be discussed.

Conclusion: lines 289-297 are not the conclusion from the results of the study. This part can be in the discussion section, but not in the conclusion.

I think that the article can be published after major revision.

Author Response

Thank you for the suggestions which improved our manuscript

General note

Please, revise the manuscript according to Checklist for Reporting Of Survey Studies (CROSS). It is advisable to include a checklist to a supplementary materials.

Accordingly, CROSS was included as supplementary file

Abstract

The results should be presented with at least some percentages instead of general phrases. Instead of the concluding phrase “Present article resumed the dentists’s percieved responsibility while using AI through the lens of: the dentist experience and qualification, prudent use of AI, and existence of conflict of interest.” there should be a brief conclusion on the results of the survey. 

Accordingly, changes were made in Abstract

Materials and methods

How was the questionnaire distributed? Was it web-based? In which language was the questionnaire? How many questions were in each section? What was the average time to complete the questionnaire?  Please, provide the results of minimum sample size calculation.

Accordingly, the information was added in the Materials and methods section. The section was rewritten.

 Results

Demographic data for the students and experienced dentists should be presented separately  in the table 1. Please, provide the calculation of the response rate for the two study groups.

Accordingly, data were presented in the text in the Results section and response rate was calculated and presented

Lines 83-84 should be explained in materials and methods section: how did you calculate Cronbah’s Alfa (between which measures?), which sample was used for this purpose

Accordingly, additional information was added in Results section.

Lines 96-100 should be moved to materials and methods section.

“According to the pre-investigation experience and the principle of the rank order scale, the basis for the classification of the three ranks is determined.” This should be further explained and the “principle of the rank order scale” should be referenced.

The principle of the rank order scale is based on the assumption that participants are able to compare the items and make relative judgments about them, even if they may not be able to provide precise or accurate ratings, therefore we used it. Reference was added and section Material and methods was rewritten.

Table 2: The scale including “very significant”, “significant”, and “non-significant” contains 2 positive and 1 negative answer. Why it was used instead of a traditional Likert scale? This should be justified in the discussion.

Accordingly, we added the following: “As these discussions revealed that there is a lack of knowledge about the subject, we used a 3-point Likert-type scale (doi:10.1177/002224377100800414) due to obtain a simple and straightforward response from participants in "forced-choice" response format. The purpose was to encourage participants to carefully consider their response and to reduce response bias that can occur when participants always select the neutral option”.

Table 3. It is better to provide exact p-values.

Figure 3 is confusing. Its idea should be explained in a clearer way.

Accordingly, we added the exact p values. Figure 3 represents how participants perceived important ethical issues related to data privacy, protection and informed consent, and responsibility of each contributor in AI use -chain.

 Discussion: 

Discussion should focus on actually discussing the results of the study. The part “Prudent use of AI in dentistry” represents a narrative review and the authors view on how ai-based systems should be used.

There is a great number of surveys on the perception of doctors and students towards artificial intelligence. These surveys and their results should be compared with the present survey in the discussion.

The limitations of the survey should be discussed.

Conclusion: lines 289-297 are not the conclusion from the results of the study. This part can be in the discussion section, but not in the conclusion.

Accordingly, we discuss results from similar studies, however, our concept was  to show how the results of the study make a wider contribution to ethical debate and practice/policy, to expand it  with ethical content and analysis, thus it has “narrative” manner. We rewrote Conclusions and add Limitations of the study.

Reviewer 3 Report

Thank you for the interesting work. Kindly find the attached PDF for comments.

Author Response

Thank you for the suggestions.

1. Accordingly, we changed the title into: “Responsible use of Artificial intelligence in dentistry: survey on dentists’ and final year-undergraduates’ perspectives”

2. Accordingly, we added more keywords

3. Accordingly, we made corrections on bar graphs, while there were already marks of pie graphs

4. We rephrased the last paragraph and put recommendations in the Discussion

Reviewer 4 Report

Dear Authors,

This is an interesting publication focused on the effectiveness of using Artificial Intelligence in dental field. The work is very interesting. However, there are several changes that need to be made for this work to be published. First of all, I suggest Authors reading carefully the Authors ‘guidelines of the Journal.  

Question 1: In all the paper sections, abbreviation should be defined at the first time. After that, use always the abbreviation.

Question 2: Introduction should be improved, in particular, the potential use of this technology  during a pandemic period, like COVID in the educational dental field.

Question 3: Authors could investigate AI to improve the application of virtual, augmented or mixed reality in Dentistry.

Question 4: Authors could explain how sample size was calculated and which parameter they used.

Question 5: Figure 2, in x axis add some reference point (as 5, 10, 15, …)

Question 6: Explain better the reason of the paragraph “Conflict of interest”

For all these reasons, I suggest a major revision before the publication.

Author Response

Thank you for the suggestions which improved our manuscript

Question 1: In all the paper sections, abbreviation should be defined at the first time. After that, use always the abbreviation.

Accordingly, after introduction, we used always abbreviations

 Question 2: Introduction should be improved, in particular, the potential use of this technology  during a pandemic period, like COVID in the educational dental field.

Accordingly, we expanded an Introduction : "AI Technology will and is already affecting educational sector, as seen in COVID-19 pandemic, aiming at providing personalized learning approach via an interactive learning experience. However, many educational institutions face challenges in effective adoption of AI into their teaching practice due to lack of teachers’ training in AI, high cost of AI software as well as ethical concerns. (doi.org/10.3390/educsci13020150; doi.org/10.3390/app12020877).  Recent introduction of new powerful AI-driven language model, ChatGPT -3, immediately showed its potential in enabling students to embrace even complex scientific concepts, but rose a number of legal and ethical concerns".

 Question 3: Authors could investigate AI to improve the application of virtual, augmented or mixed reality in Dentistry. And Question 4: Authors could explain how sample size was calculated and which parameter they used.

Accordingly, Material and methods section was rewritten. Since this survey aimed to assess dentists’ and final year-students’ familiarity with, and attitudes toward AI use in dentistry, without any specific hypothesis testing, sample size estimation and power calculation were waived.  Limitations of the study were emphasized as well as need for future research to expand on the application of virtual, augmented or mixed reality in Dentistry.

Question 5: Figure 2, in x axis add some reference point (as 5, 10, 15, …)

Thank you for the remark. Accordingly, we have made the corrections

 Question 6: Explain better the reason of the paragraph “Conflict of interest”

Conflict of interest relates to situations which often involve professionally related commercial enterprises or financial arrangements in which the individual is involved and personally benefits. Thus, conflict of interest is related to situation when dentist’s participation, due to financial gain,  creates bias that could adversely influence the decision.  In order to be more clear about the importance of this issue for the matter of dentist’s accountability while using AI, we rearranged the place of this paragraph in the manuscript.

Round 2

Reviewer 2 Report

Dear authors,

you’ve made sufficient improvements and the article is better now. No sample size calculation should be mentioned in the limitations.

Author Response

Thank you for suggestion. We modified limitations of the study.

Reviewer 4 Report

Dear Auhtors

You have responded to all comments. However, I decided on a minor revision because references should be added regarding lines 38-40. For example, how AI could help the clinician understand the prevalence, characteristics, and degree of association of oral lesions in COVID-19. Also lines 42-45 need references. 

Author Response

Thank you for suggestions. References are added accordingly.